# Impact of social support on PTSD : Chain mediating effects of insomnia and anxiety

Xiaofei Mao[1], Tianya Hou[2] , Yulin Zhang[3,4], Jianguo Zhang[2], Fan Zhang[1] and Weizhi Liu[1]

[1]Lab for Post-traumatic Stress Disorder, Faculty of Psychology, Naval Medical University, Shanghai, China; [2]Faculty of Psychology, Navy Medical University, Shanghai, China; [3]National Key Laboratory of Human Factors Engineering, Beijing, China and [4]China Institute of Marine Technology and Economy, Beijing, China

**Keywords:**
social support; insomnia; anxiety; PTSD; COVID-19; nurses

**Corresponding author:**
Weizhi Liu;
Email: 13024141970@163.com

X.M. and T.H. share joint first authorships equally.

## Abstract

**Objective:** To reveal the chain mediating roles of insomnia and anxiety between social support and PTSD in nursing staff under the stage of COVID-19 regular pandemic prevention and control in China.
**Methods:** A total of 784 nurses were recruited using the convenience sampling method in Jiangsu Province, China. Demographic questionnaire, Perceived Social Support Scale, Impact of Event Scale-Revised, Generalized Anxiety Disorder-7 and Insomnia Severity Index were applied to collect data.
**Results:** Social support, PTSD, insomnia and anxiety were significantly correlated with each other. Insomnia and anxiety acted as chain mediators between social support and PTSD.
**Conclusion:** Insufficient social support may trigger PTSD through the chain mediating effects of insomnia and anxiety in nursing staff under the stage of COVID-19 regular pandemic prevention and control. Measures focusing on social support, insomnia and anxiety should be taken to reduce or even prevent PTSD in nursing staff in Chinese hospitals in similar crises in the future.

## Impact statement

Chinese nurses played a vital role in combating the virus under great pressure and during the COVID-19 regular pandemic prevention and control. It's of great importance to investigate effects of insomnia and anxiety between social support and PTSD among Chinese nurses in the context of regular COVID-19 pandemic prevention and control. It could aid the hospital with interventions to ameliorate PTSD among Chinese nursing staff during COVID-19 regular pandemic prevention and control. However, studies focusing on the relationship among these variables are insufficient. The present study aimed to focus on how social support would influence PTSD in Chinese nursing staff during the period of COVID-19 regular pandemic prevention and control.

## Introduction

The coronavirus disease 2019 (COVID-19) has brought about mental problems to the healthcare staff (Chew et al., 2020a; Chew et al., 2020b; Le et al., 2021; Pham et al., 2021; Tan et al., 2023), especially nurses (Dong et al., 2022). Studies suggested that the nursing staff suffered more problems from trauma-related stress during the outbreak of COVID-19 (Benfante et al., 2020; Chatzittofis et al., 2021; Chen et al., 2021a; Yunitri et al., 2022). According to recent literature in China, the incidence rate of PTSD in Chinese nurses following the outbreak of COVID-19 was 36.2% (Lai et al., 2020).

China stepped into the stage of the COVID-19 regular pandemic prevention and control on May 7, 2020 (The State Council, 2020). Therefore, the policy of national COVID-19 pandemic prevention and control was changed into regular pandemic prevention and control (Xinhua Press, 2020), resulting in an increased workload of nurses.

With the COVID-19 epidemic prevention work changing from emergency to normalization, hospitals were checkpoints for epidemic prevention and control, confronting potential risk factors and the nosocomial infection prevention and control measures will exist for a long period of time (Wu et al., 2022). The contents of nurses' jobs have changed. Excluding routine nursing work, they need to take on more responsibilities, including the strict procedure of admission of patients to hospitals and nursing services for patients from high- and medium-risk areas (Jiang et al., 2022). Moreover, constantly working under stressful environments has also resulted in pandemic fatigue among nurses (Zhang et al., 2021). Thus, though the grim situation faced by

Chinese nurses has changed, they still work under great pressure under the stage of COVID-19 regular pandemic prevention and control.

About 16.10% of Chinese nurses from low-risk areas still experienced moderate to severe psychological pressure during the stage of COVID-19 regular pandemic prevention and control (Chen et al., 2021b). Moreover, Li and colleagues found that 50.54% of the frontline nurses in Wuhan, China, reported moderate (42.21%) to severe (8.33%) levels of stress in the context of COVID-19 regular pandemic prevention and control (Wen et al., 2022). Nurses' mental health was directly related to their performance quality in caring for patients and work efficiency (Mirzaei et al., 2022). PTSD was an important aspect of psychological health in Chinese nursing staff exposed to COVID-19 (Lai et al., 2020). Therefore, it's of great importance to investigate the influential factors of PTSD in Chinese nursing staff in the context of regular COVID-19 pandemic prevention and control.

Social support is defined as "the feeling that one is cared for and has assistance available from other people" and "that one is part of a supportive social network" (Somville et al., 2016). The stress-buffering hypothesis supposes that social support plays a role in buffering stress by increasing the levels of self-efficacy and self-esteem (Cohen et al., 1985). Perceived social support was positively correlated with mental well-being and could help cultivate the ability to overcome stressful events (Xu et al., 2019; Huang et al., 2020). According to existing literature focusing on PTSD, social support was thought to be a strong predictive factor of PTSD among adults. A meta-analysis focusing on risk factors of PTSD has shown that low level of social support was the strongest predictor of PTSD (Brewin et al., 2000). Similarly, another meta-analysis conducted by Ozer and colleagues suggested a similar result (Ozer et al., 2003). Moreover, Guay et al. reviewed numerous studies and summarized that social support played a significant role in PTSD (Guay et al., 2006).

Recently, PTSD among medical staff has drawn plenty of attention from researchers. For example, it was supposed that lacking social support was a significant risk factor for PTSD among traumatized nurses from South Korea (Kim and Yeo, 2020). Besides, lacking social support was found to be a vital predictive factor of PTSD in healthcare staff fighting against the COVID-19 pandemic (d'Ettorre et al., 2021).

Therefore, we could infer that those lacking social support might be more prone to PTSD among nurses under the stage of COVID-19 regular pandemic prevention and control in China.

Insomnia has been proven to be associated with PTSD of healthcare staff during the outbreak of COVID-19. Blekas and colleagues found that 73.3% of Greek healthcare professionals with PTSD reported problems of insomnia. However, only 28% of the Greek healthcare professionals without PTSD suffered from bad sleep quality (Blekas et al., 2020). Yin et al. conducted a structural equation model and found that the sleep quality of Chinese medical staff could significantly predict PTSD symptoms (Yin et al., 2021). Li et al. discovered that insomnia played a significant partial mediating role between being a nurse and post-traumatic stress symptoms during the outbreak of COVID-19 (Li et al., 2022).

A study of Japanese daytime workers revealed that social support was an independent risk factor for insomnia (Nakata et al., 2004). Besides, social support was significantly related to the increased risk of sleeping difficulties during the COVID-19 pandemic. Specifically, people with good social support reported lower risk of sleeping difficulties (Grey et al., 2020). Moreover, it could be inferred from the existing literature that the protective effect of

support from organizations on sleep quality among Chinese healthcare workers (Zou et al., 2021) and nurses (Du et al., 2022) during the outbreak of COVID-19.

Based on the studies above, insomnia may play a mediating role between social support and PTSD in Chinese nursing staff during COVID-19 regular pandemic prevention and control.

Similarly, anxiety was also found to be a predictive factor of PTSD among medical professionals during the COVID-19 pandemic (Lu et al., 2021; Yin et al., 2021; Yao et al., 2022). For example, Lu et al. found anxiety symptoms of COVID-19 frontline medical staff from Taiwan, China, significantly predicted PTSD using a method of hierarchical linear regression model (Lu et al., 2021). Moreover, social support had a positive effect on anxiety among 4,658 adults (Roohafza et al., 2014). Another research during the COVID-19 pandemic found that sufficient social support might reduce anxiety in medical staff (Zhu et al., 2020).

Hence, it is reasonable to infer that anxiety may mediate the relationship between social support and PTSD in Chinese nursing staff during COVID-19 regular pandemic prevention and control.

Insomnia and anxiety always co-occurred and were significantly associated (Jansson-Fröjmark and Lindblom, 2008; Marcks and Weisberg, 2009; Chinese Society of Neurology, 2020). Plenty of research investigated the relationship between insomnia and anxiety. Taylor et al. discovered that insomnia played a negative role in the elevated risk for anxiety in their review (Taylor et al., 2003). Besides, subjects suffering from sleeping difficulties reported more symptoms of anxiety than those with good sleep quality (Taylor et al., 2005). Moreover, participants diagnosed with insomnia suffered from higher levels of anxiety, indicating the vital role of insomnia in anxiety (Mason and Harvey, 2014; Chen et al., 2017). These studies above indicated that insomnia might lead to anxiety.

Consequently, it could be inferred that insufficient social support might lead to PTSD. Besides, insomnia and anxiety were significant mediators between social support and PTSD. Chain mediating model refers to a condition that there are two or more mediators in a mediating model, which is used to study how an independent variable influences dependent variable through a series of variables. In the chain mediation model, the mediators are not independent of each other (Liu and Ling, 2009). Given existing evidence about the effect of insomnia on anxiety, we supposed that insomnia and anxiety might act as chain mediators in the association between social support and PTSD among nurses during COVID-19 regular pandemic prevention and control.

Therefore, the current study aimed to reveal the associations between social support, insomnia, anxiety, and PTSD among nurses. We hypothesized that insomnia and anxiety played chain mediating roles between social support and PTSD in nursing staff during the stage of COVID-19 regular pandemic prevention and control in China. Our research is of realistic significance and could aid the hospital with interventions to ameliorate PTSD among Chinese nursing staff during COVID-19 regular pandemic prevention and control. A chain mediation model was constructed to test our hypotheses (Figure 1). The variables in Figure 1 were listed as follows: X = Social support, Y = PTSD, M1 = Insomnia, M2 = Anxiety. Specifically, social support was the independent variable, and PTSD was the dependent variable. Insomnia and anxiety were calculated as chain mediators in the relationship between social support and PTSD among nursing staff.

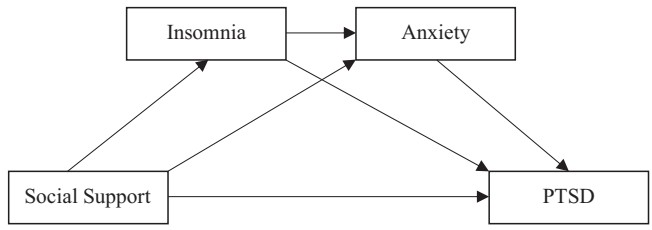

**Figure 1.** Hypothesized model about social support, insomnia, anxiety, and PTSD.

## Methods

### Participants and procedures

The current research was conducted in January 2022. Inclusion criteria: I) normal ability of speech, comprehension, and expression; II) ≥18 years old; III) working as a nurse during COVID-19 regular pandemic prevention and control. Exclusive criteria: diagnosed with psychiatric illness before. A total of 784 nurses were recruited from five tertiary hospitals using a convenience sampling method in China.

The mean age of the participants was 26.37 years with a standard deviation of 6.59. 94.4% of the nurses were female. About two in three of the respondents were married. The average working seniority was 8.81 ± 6.87 years. A detailed flowchart is shown in Figure 2.

### Measures

### Demographics

Demographic information (e.g., age, gender, marital status, and working seniority) of nurses was collected in this study.

### Perceived social support scale (PSSS)

PSSS includes 12 items and is a classic evaluation tool for measuring the perceived social support (Zimet et al., 1988). Each item is rated from 0 (strongly disagree) to 7 (strongly agree). The highest score of the scale is 84. The scale has shown good validity and reliability and has been widely used among the Chinese

population (Mao et al., 2023a; Mao et al., 2024). In this study, the coefficient of Cronbach's alpha was 0.968.

### Insomnia severity index (ISI)

ISI is a 7-item questionnaire and a classic evaluation tool for perceived insomnia severity (Bastien et al., 2001). Each item was rated from 0 (not at all) to 4 (nearly every day). The highest score of the scale is 28. The scale has presented sufficient validity and reliability and has been widely used among the Chinese population (Mao et al., 2023b; Mao et al., 2024). In this study, the coefficient of Cronbach's alpha was 0.927.

### Generalized anxiety disorder-7 (GAD-7)

GAD-7 is a 7-item questionnaire and a classic evaluation tool for anxiety and its severity (Spitzer et al., 2006). Participants were asked to report the frequency of each item during the last 2 weeks. "Not at all," "several days," "more than half the days" and "nearly every day" were scored as 0, 1, 2, and 3, respectively. The highest score of the scale is 21. The scale has been widely used with excellent validity and reliability among the Chinese population (Mao et al., 2023c; Mao et al., 2024). In this study, the coefficient of Cronbach's alpha was 0.960.

### Impact of event scale-revised (IES-R)

IES-R is a 22-item and a classic evaluation tool for assessing post-traumatic stress symptoms. There are three subscales (intrusiveness, avoidance, and hyperarousal) in this scale. The highest score of the scale is 88. The cut-off of IES-R is 33 (Asif et al., 2016). The scale has shown good validity and reliability in the Chinese population (Mao et al., 2023d; Mao et al., 2024). In this study, the coefficient of Cronbach's alpha was 0.976."

### Statistical analysis

IBM SPSS (version 21.0) was adopted to analyze the data. The significance level is set as α = 0.05, and all tests are double-tailed tests (Mao et al., 2024). Common method bias was detected by the Harman single-factor test (Mao et al., 2023b). Hence, the Harman

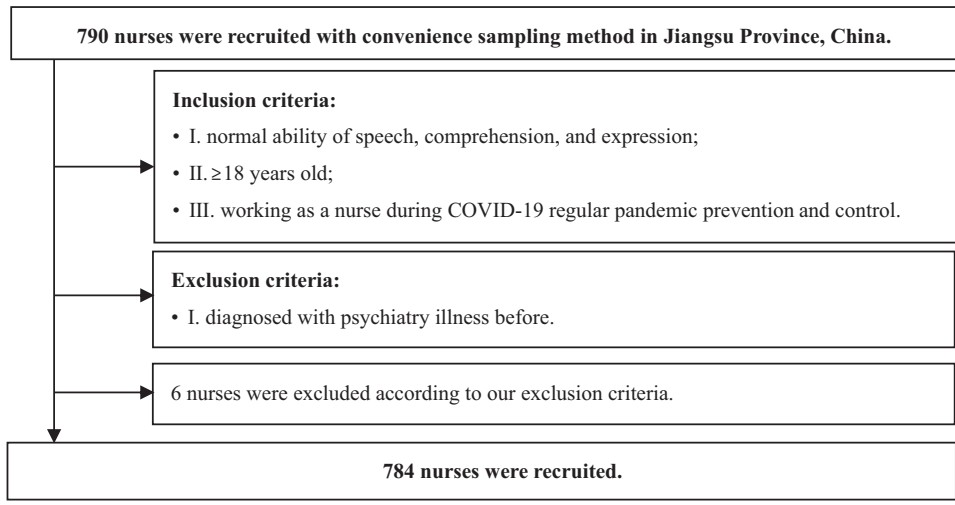

**Figure 2.** Flowchart of subjects' enrollment in January 2022.

single-factor test was chosen to detect the presence of common method bias in this study. Pearson correlation analyses is a common statistical method to detect the correlations between variables (Hou et al., 2021). Therefore, we calculated the correlations among anxiety, insomnia, social support, and PTSD with Pearson correlation analyses. The Hayes SPSS macro program process (version 3.4.1, Model 6) is a good tool to detect the chain mediating model (Hayes, 2013). We adopted the Hayes SPSS macro program process (version 3.4.1, Model 6) to conduct the bootstrapped method (5000 resamples) to estimate 95% confidence interval (CI) for significance testing for mediating model (Preacher and Hayes, 2008). According to the method, the chain mediation model would be established if zero was included in 95% CI.

## Results

### Common method Bias test

Common method bias may appear in self-report inventory (Campbell and Fiske, 1959). Harman single-factor test was selected to detect if common method bias existed in this study (Podsakoff et al., 2003; Zhou and You, 2004). Six values with an eigenvalue more than 1 were found. The first factor accounted for a variance of 36.81% (<40%). Therefore, common method bias did not exist in the current study.

### Pearson's correlation

Results of Pearson's correlation among social support, insomnia, anxiety, and PTSD (Table 1) indicated that four variables were related to each other significantly (all $P < 0.01$).

### Mediating roles of insomnia and anxiety

PROCESS 3.4.1 (Model 6) was applied to test the chain mediating model (Figure 3). Studies revealed age, gender, marital status (Zhou et al., 2021) and working seniority (Goh et al., 2021) were closely related to PTSD. Therefore, demographic information (age, gender, marital status, and working seniority) was controlled as confounders.

As shown in Table 2, the direct effect was significant since the 95% CI ($-0.245 \sim -0.102$) did not contain 0. The total mediating effect was significant since 0 wasn't included in the 95% CI ($-0.382 \sim -0.260$). The indirect effect of social support on PTSD via insomnia was significant as 0 wasn't contained in the 95% CI ($-0.064 \sim -0.012$). Similar results were detected among the mediating effects of anxiety and the chain mediating effects of insomnia and anxiety.

Figure 3 shows the chain mediating effects of insomnia and anxiety in the association between social support and PTSD. As showed in Figure 3, all paths in chain mediating model were significant (**$P < 0.01$, ***$P < 0.001$). The results revealed that

**Table 1.** Results of bivariate analysis among variables

| Variables | M ± SD | 1 | 2 | 3 | 4 |
|---|---|---|---|---|---|
| Social support | 61.73 ± 12.70 | 1 | | | |
| Insomnia | 6.31 ± 5.27 | −0.29** | 1 | | |
| Anxiety | 3.62 ± 4.06 | −0.39** | 0.44** | 1 | |
| PTSD | 17.31 ± 16.09 | −0.38** | 0.39** | 0.67** | 1 |

Note: **$P < 0.01$.

the negative relationship between social support and PTSD was significant, with insomnia and anxiety partially mediating the relationship between social support and PTSD, respectively. Besides, a significant chain mediating effect of insomnia and anxiety was also detected in the current research. The total mediating effects was 64.77%.

To sum up, insomnia and anxiety played chain mediating roles between social support and PTSD among nurses during COVID-19 regular pandemic prevention and control.

## Discussion

The present investigated how social support would influence PTSD among nurses during the period of COVID-19 regular pandemic prevention and control in China. The results revealed that insomnia and anxiety played chain mediating roles between social support and PTSD. As far as we know, our study is the first to focus on the effect of social support on PTSD among Chinese nursing staff via chain mediating effect of insomnia and anxiety.

We discovered social support could negatively and significantly predict PTSD, which was in line with previous research. Schnurr and colleagues investigated influencing factors of PTSD among Vietnam veterans. They found that social support played an important role in the development and maintenance of PTSD (Schnurr et al., 2004). Besides, leadership social support was supposed to be related to PTSD in pediatric oncology nursing staff (Schuster et al., 2022). Song et al. conducted a cross-sectional study on 14,825 Chinese medical staff during the outbreak of COVID-19. They found that those with insufficient social support were more likely to develop PTSD (Song et al., 2020).

Social support is a protective factor against psychological difficulties. Kerasiotis et al. suggested that social support might act as a protective factor in helping nursing staff to deal with work pressure (Kerasiotis and Motta, 2004). The coping theory supposed that social support was considered as a vital coping strategy when facing stress. Specifically, social support might have a significant effect in relieving negative influence from kinds of pressure events by solving problems and relieving negative emotions via assistance from social ties (Mo et al., 2020). Besides, the stress-buffering hypothesis supposes that social support plays a role in buffering stress by increasing the levels of self-efficacy and self-esteem (Cohen et al., 1985). Insufficient social support played a predictive role in PTSD among medical staff during the COVID-19 pandemic (d'Ettorre et al., 2021). A Korean study indicated that nurses might experience more PTSD if they got lower levels of support from their managers during the COVID-19 pandemic (Bae et al., 2022). Moreover, social support was thought to be a protective factor of psychological resilience that maintained mental health and eliminated psychological barriers (Sripada et al., 2016). This may be the possible explanation for the positive effect of social support on PTSD.

As hypothesized, we demonstrated the chain mediating roles of insomnia and anxiety. Social support could influence PTSD through three paths (Figure 3). First, paths 1 and 2 indicated that insomnia and anxiety played significant mediating roles between social support and PTSD, respectively. The results indicated close relationships among insomnia, anxiety, and PTSD. Specifically, insomnia and anxiety were positively predictive factors of PTSD among nurses during COVID-19 regular pandemic prevention and control, which were in line with previous studies. For example, Yao et al. found sleep disorder and anxiety were significant mediators of PTSD in medical staff during the COVID-19 pandemic (Yao et al.,

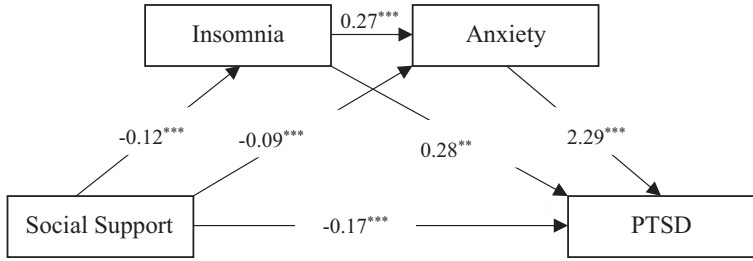

**Figure 3.** Chain mediating model about social support, insomnia, anxiety, and PTSD (**P < 0.01, ***P < 0.001).

**Table 2.** Chain mediating roles of insomnia and anxiety

| | Path | Effect value | Effect size | 95% CI | |
|---|---|---|---|---|---|
| | | | | LLCI | ULCI |
| Mediating effects | 1: Social support→Insomnia→PTSD | −0.035 | 7.13% | −0.064 | −0.012 |
| | 2: Social support→Anxiety→PTSD | −0.207 | 42.16% | −0.105 | −0.055 |
| | 3: Social support→Insomnia→Anxiety→PTSD | −0.076 | 15.48% | −0.275 | −0.149 |
| Direct effect | Social support→PTSD | −0.173 | — | −0.245 | −0.102 |
| Total mediating effect | | −0.318 | 64.77% | −0.382 | −0.260 |
| Total effect | | −0.491 | — | −0.574 | −0.409 |

2022). Similarly, Yin et al. found sleep quality and anxiety were mediators of PTSD in medical staff combating against COVID-19 (Yin et al., 2021). Besides, Li and colleagues discovered that insomnia was a significant partial mediator, which explained 32.53% of the association between being a nurse and post-traumatic stress symptoms among frontline medical staff during the COVID-19 pandemic in China (Li et al., 2022). Meanwhile, anxiety symptoms of COVID-19 frontline healthcare workers from Taiwan, China were proved to be a significant predictor of PTSD (Lu et al., 2021). Moreover, social support was proven to be a predictive factor of insomnia and anxiety among Turkish medical professionals (Yılmaz et al., 2021). The same results were also observed in Chinese medical staff. Social support was negatively related to sleep quality and anxiety significantly (Xiao et al., 2020). Secondly, path 3 suggested that insomnia and anxiety acted as chain mediators in our model. Namely, insufficient social support would lead to sleep problems, which may increase the level of anxiety and then result in PTSD among nurses. However, previous studies did not classify the relationship among social support, insomnia, anxiety, and PTSD.

The current research is the first study to investigate possible mechanisms underlying the relationship between social support and PTSD of nurses during COVID-19 regular pandemic prevention and control.

## Implications

The current study has several implications for the amelioration of PTSD in nursing staff after public health emergencies. Our study gave empirical evidence for the possible mechanism underlying the relationship between social support and PTSD among Chinese nursing staff. Social support could not only directly influence PTSD, but also indirectly impact PTSD through three pathways: (1) social support-insomnia-PTSD; (2) social support-anxiety-PTSD; (3) social support-insomnia-anxiety-PTSD. The results of

our research suggested to the Chinese hospital authorities the possible methods to reduce or even prevent PTSD. The nursing staff should take action concentrating on promoting sleep problems and anxiety. Prior studies revealed that cognitive behavioral therapy (CBT) and internet cognitive behavioral therapy (I-CBT) were effective in treating psychiatric symptoms (Zhang and Ho, 2017) or psychological outcomes during the COVID-19 pandemic (Ho et al., 2020; Soh et al., 2020). Therefore, CBT and I-CBT focusing on promoting sleep quality and anxiety could be taken to reduce PTSD among nurses.

## Limitations

There are some limitations in our study. Firstly, the cross-sectional design adopted in the current study could not determine causal associations among variables. Longitudinal or experimental approaches are needed in the future to infer the possible causal relationships among variables. Second, our subjects were recruited using the convenience sampling method. The small sample size might limit the sample representativeness, which may limit the generalizability of the findings. Further research had better-recruited subjects with random cluster sampling and expanded the sample size to enhance the external validity. Thirdly, burnout during the COVID-19 pandemic is an important public health issue and is worthy of further study (Mao et al., 2024; Lau et al., 2022). However, burnout was not considered in the present study. Future research should consider this issue to explore the association of burnout with PTSD among nurses.

## Conclusion

To conclude, the study demonstrated the chain mediating roles of insomnia and anxiety in the relationship between social support and PTSD among nursing staff during COVID-19 regular

pandemic prevention and control. The Chinese hospital authorities need to pay extra attention to nursing staff with insufficient social support to prevent and reduce PTSD. Moreover, actions targeting the reduction of insomnia and anxiety also need to be taken to reduce or even prevent PTSD in nursing staff.

**Open peer review.** To view the open peer review materials for this article, please visit http://doi.org/10.1017/gmh.2024.91.

**Data availability statement.** The data that support the findings of this study are available on request from the corresponding authors. The data are not publicly available due to the de-identified data possibly containing information that could compromise the privacy and safety of the research participants.

**Acknowledgements.** The authors would like to acknowledge the volunteers who participated in the study.

**Author contribution.** Xiaofei Mao, Tianya Hou, and Weizhi Liu designed the study. Yulin Zhang, Jianguo Zhang, and Fan Zhang collected the data. Xiaofei Mao and Tianya Hou analyzed the data and drafted the manuscript. Weizhi Liu led the whole study. All the authors have read and approved the final manuscript.

**Financial support.** The authors disclosed receipt of the following financial support for the research, authorship, and/or publication of this article: Youth Initiation Fund of Naval Medical University (2023QN028); Social Science Cultivation Fund of Naval Medical University (2023SK015); Youth Talent Initiation Fund of Faulty of Psychology of Naval Medical University (2023RC003). Youth Social Science Cultivation Fund of Faulty of Psychology of Naval Medical University (2023SK001).

**Competing interest.** The authors declare no competing interests.

**Ethics statement.** All participants signed a written informed consent form in ethics approval and consent to participate. The study was approved by the ethics committee of Naval Medical University in accordance with the ethical standards established in the 1964 Declaration of Helsinki and its later amendments.

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
