## [Editor Report]

Dear authors, the team of reviewers and I have reviewed the article very carefully, we have comments and we need your revisions and that you send us a new version to continue with the process. We send you the comments below and ask you to respond as soon as possible:

Reviewer 1: 

The title effectively piqued my interest, and the overall topic of the article aligns well with current research. However, the reporting of the results could benefit from some improvements.

Current problems:

1. Most of the parts are NOT aligned with Author guidelines.

2. Introduction was written in separated section rather than in a seamless flow.

3. The author described their Methods in details; unfortunately, the formality should be revised to align with journal’s guidelines.

4.To strengthen the connection between the results and their interpretations, the authors could consider reorganizing the evidence for a clearer flow and provide a more nuanced discussion of the implications.

With these revisions, the article would be well-suited for submission to a high-quality journal (Q3-Q4). The authors should not be demotivated with these comments and I wish them all the best with future endeavors.

Reviewer 2: 

I have the following comments for the authors to address and I am happy to review this paper again.

1) In the introduction, please discuss the following important research findings:

Search PubMed for: They also perceived the workplace response to COVID-19 as inadequate (score = 2.0, SD = 0.5). In particular, participants who worked in the emergency or intensive care departments were more likely to perceive an exposure risk, compared to those in infectious disease control departments.

Search PubMed for: Nurses had relatively higher percentages of having normal or unchanged psychological distress symptoms relative to other health care workers (n=233-260 [86.0%-95.9%] vs n=187-199 [74.8%-91.7%]). Among those without psychological symptoms, nurses constituted a higher proportion than doctors and nonmedical health care workers (n=194 [40.2%], n=142 [29.5%], and n=146 [30.3%], respectively).

Search PubMed for : In the multivariable analysis, we found that non-medically trained personnel, the presence of physical symptoms and presence of prior medical conditions were independent predictors across the participating countries.

Search PubMed for: a significant association between the prevalence of physical symptoms and psychological outcomes among healthcare workers during the COVID-19 outbreak. We postulate that this association may be bi-directional, and that timely psychological interventions for healthcare workers with physical symptoms should be considered once an infection has been excluded.

Search PubMed for: Being women also had a negative association with scores in “Avoidance of disclosure and discrimination related to COVID 19” domain (Coef. = -0.27, CI: -0.43 to -0.12) while having a positive association with “negative attitude towards working conditions” domain (Coef. = 0.19, CI: 0.09 to 0.3). In addition, working in administrative offices (Coef. = 0.20; 95% CI = 0.05 to 0.36) and infectious departments (Coef. = 0.36; 95% CI = 0.09 to 0.63) had a positive association with “Increased work pressure due to COVID 19” domain.

Search PubMed for: Out of the 906 healthcare workers who participated in the survey, 48 (5.3%) screened positive for moderate to very-severe depression, 79 (8.7%) for moderate to extremely-severe anxiety, 20 (2.2%) for moderate to extremely-severe stress, and 34 (3.8%) for moderate to severe levels of psychological distress. The commonest reported symptom was headache (32.3%), with a large number of participants (33.4%) reporting more than four symptoms.

Search PubMed for: Respondents who knew of someone diagnosed with COVID-19 were more likely to screen positive for depression, stress, and PTSD (OR 1.2, 1.2, and 1.3 respectively, all P < 0.05). Surgical specialties that operated in the head and neck region had higher psychological distress among its surgeons.

2) Under discussion, please discuss the psychological interventions such as Internet CBT to help nurses based on the following findings:

The most evidence-based treatment is cognitive behaviour therapy (CBT), especially Internet CBT that can prevent the spread of infection during the pandemic.

Use of Cognitive Behavior Therapy (CBT) to treat psychiatric symptoms during COVID-19:

Mental Health Strategies to Combat the Psychological Impact of COVID-19 Beyond Paranoia and Panic. Ann Acad Med Singapore. 2020;49(3):155‐160.

Cost-effectiveness of iCBT:

Moodle: The cost effective solution for internet cognitive behavioral therapy (I-CBT) interventions. Technol Health Care. 2017;25(1):163-165. doi: 10.3233/THC-161261. PMID: 27689560.

Internet CBT can treat psychiatric symptoms such as insomnia:

Efficacy of digital cognitive behavioural therapy for insomnia: a meta-analysis of randomised controlled trials. Sleep Med. 2020 Aug 26;75:315-325. doi: 10.1016/j.sleep.2020.08.020. Epub ahead of print. PMID: 32950013.

3) Under discussion, please discuss the limitation for lack of measuring COVID-19 burnout based on the following study:

Search PubMed for: Burnout is an important public health issue at times of the COVID-19 pandemic. Current measures which focus on work-based burnout have limitations in length and/or relevance. When stepping into the post-pandemic as a new Norm Era, the burnout scale for the general population is urgently needed to fill the gap. This study aimed to develop a COVID-19 Burnout Views Scale (COVID-19 BVS) to measure burnout views of the general public in a Chinese context and examine its psychometric properties.

Reviewer 3: 

Introduction:

- The introduction lacks clarity regarding the objectives of the study, failing to explicitly state the aim of investigating the relationship between social support, insomnia, anxiety, and PTSD among nursing staff.

- Limited elaboration on the specific challenges faced by nursing staff during the regular pandemic prevention and control phase, reducing the contextual understanding of the study’s relevance and significance.

- Consider providing a brief overview of the theoretical framework guiding the study to enhance understanding for readers.

Methods:

- Reliance on convenience sampling introduces potential bias and limits the generalizability of the findings.

- The cross-sectional design restricts the ability to establish causal relationships between variables, suggesting the need for alternative designs such as longitudinal or experimental approaches.

- Clarify the rationale behind the choice of specific measures and methods used in the study to justify their appropriateness for the research questions.

Results:

- Absence of discussion on potential confounding variables controlled for in the analysis.

- Relatively small sample size may impact the robustness of the findings.

Provide additional context for the interpretation of results by discussing effect sizes and practical significance, not just statistical significance.

Discussion:

- Inadequate discussion of study limitations, including the cross-sectional design and sampling method.

Suggestions for future research could be more expansive to address identified limitations and gaps in the literature.

- Consider integrating implications for clinical practice and policy interventions more explicitly to enhance the practical relevance of the findings.

Conclusion:

- Limited exploration of alternative explanations or factors influencing the observed relationships.

- Lack of nuanced interpretations of findings and their implications for clinical practice and policy interventions.

- Provide a more comprehensive discussion of the theoretical implications of the findings and their alignment with existing literature in the field.

Reviewer4: 

This study aims to examine the social support on PTSD of nursing staff during COVID-19 regular pandemic prevention and control in China. There are some follows concerns:

1, it is hard to test the mediate effects through the cross section data.

2, The introduction section is too weak, and lack some improtant references related this issues. Also, it is lack the theory evidence for the chain mediation effects.

3. The logic of the whole text needs to be further sorted out。

4. The sampling is not very clear. For example, from how many hospitals, what type of hospital, etc.

5. Data analysis is inadequate. 

Reviewer5: 

Details of the PROCESS 3.4.1 (Model 6) could be elaborated for better understanding

Statistics behind figure 2 and figure 3 needs to be explained in detail

---

## [Editor Report]

Dear authors, thank you very much for reviewing the article, we return the pampering with very few suggestions.

Review 1

Good edits were incorporated for revisions.

Minor suggestions for improvement:

1) keyword - nursing (plural) instead of nurse (singular)

2) “corelated” should be corrected to “correlated”.

3) “Dec” should be written as “December” in “The current research was conducted in December 2022.”

4) In “Exclusive criteria: diagnosed with psychiatry illness before.”, “psychiatry” should be changed to “psychiatric.”

5) In “The mean age of the participants was 26.37±6.59 years old.”, it is recommended to specify which value is the mean and which is the standard deviation to avoid ambiguity. For example, “The mean age of the participants was 26.37 years with a standard deviation of 6.59.”

6) In “The stress-buffering hypothesis supposed that social support played a role in buffering stress via increasing the levels of self-efficacy and self-esteem.”, “supposed” should be “supposes.”

7) “interpret to” should be revised to “infer from” in “Moreover, existing evidence reveal the protective effect of support from organizations to sleep quality among Chinese healthcare workers.”

8) In “Hyphothesized model about social support, insomnia, anxiety, and PTSD.”, “Hyphothesized” should be corrected to “Hypothesized.”

9) “Figure 2. Flowchart of subjects’ enrollment.” should have a period at the end to make it consistent with other figure captions.

10) “Harman single factor test was selected to detect if common method bias existed in this study.” could be revised to “Harman single-factor test was chosen to detect the presence of common method bias in this study.”

11) “chain mediating model” could be clarified to avoid confusion. It may benefit from further explanation or a brief definition. What is chain mediating model?

12) The statement “However, burnout wasn’t involved in the present study.” could be rephrased as “However, burnout was not considered in the present study.”

13) “To sum up,” could be changed to “In conclusion,” or “To conclude,” for variety.

14) In the last sentence of the conclusion paragraph, consider revising “actions focusing on the improvement of insomnia and anxiety also need to be taken” for better clarity. 

Thank you!